# Women's health in *The BMJ*: a data science history

Eva N Hamulyák  ,[1] Austin J Brockmeier,[2] Johanna D Killas,[3] Sophia Ananiadou,[4,5] Saskia Middeldorp,[1] Armand M Leroi[6,7]

SM and AML are joint senior authors.

¹Department of Vascular Medicine, Amsterdam UMC, University of Amsterdam, Amsterdam, The Netherlands
²Department of Electrical & Computer Engineering, University of Delaware, Newark, Delaware, USA
³Health Studies Programme, University of Toronto, Toronto, Ontario, Canada
⁴Department of Computer Science, The University of Manchester National Centre for Text Mining, Manchester, UK
⁵The Alan Turing Institute, London, UK
⁶Department of Life Sciences, Imperial College London, South Kensington Campus, London, UK
⁷Data Science Institute, Imperial College London, South Kensington Campus, London, UK

**Correspondence to**
Dr Eva N Hamulyák;
e.n.hamulyak@amsterdamumc.nl

## ABSTRACT

**Objective** To determine how the representation of women's health has changed in clinical studies over the course of 70 years.

**Design** Observational study of 71 866 research articles published between 1948 and 2018 in *The BMJ*.

**Main outcome measures** The incidence of women-specific health topics over time. General linear, additive and segmented regression models were used to estimate trends.

**Results** Over 70 years, the overall odds that a word in a *BMJ* research article was 'woman' or 'women' increased by an annual factor of 1.023, but this rate of increase varied by clinical specialty with some showing little or no change. The odds that an article was about some aspect of women-specific health increased much more slowly, by an annual factor of 1.004. The incidence of articles about particular areas of women-specific medicine such as pregnancy did not show a general increase, but rather fluctuated over time. The incidence of articles making any mention of women, gender or sex declined between 1948 and 2005, after which it rose steeply so that by 2018 few papers made no mention of them at all.

**Conclusions** Over time women have become ever more prominent in *BMJ* research articles. However, the importance of women-specific health topics has waxed and waned as researchers responded ephemerally to medical advances, public health programmes, and sociolegal changes. The appointment of a woman editor-inchief in 2005 may have had a dramatic effect on whether women were mentioned in research articles.

## INTRODUCTION

Sex matters in medicine.[1] An unequal representation of women exists in leadership and medicine, and women-specific topics are often devalued.[2] From a health perspective, women and men differ in their reproductive biology, but also in the risks of many non-reproductive diseases such as autoimmune disorders and venous thromboembolism,[3 4] the relative importance of disease risk factors,[5 6] rates of diagnosis, prognoses, and how they respond to drugs.[7 8]

Compared with men, women have historically been and, in some ways still are, ill served by medicine.[5] The deficiencies of early medicine in the treatment of women

are well documented,[9–12] and so are the 20th century's campaigns for change.[13–17] The history of women's health in modern—post-1900—clinical science has, however, been little studied. An exception is the fraught discussion over whether women have been adequately represented, or properly studied, in clinical trials.[18–23] The tendency to use men as the standard in clinical research, driven by concerns for potential teratogenic effects of drugs or by deeming women's inclusion as risker, have been suggested as explanations for the under-representation of women in trials.[5] Even if both sexes were included, sex disaggregation was not performed, as it is becoming standard now.[21]

Clearly, sex-specific differences are present and must be considered in clinical decision making, but the question of how women have been studied by clinical science is a much larger one than this. It includes the focus and emphasis of medical research, the communication of these findings and their translation to clinical practice, all of which have consequences for health policy.[1]

Here we apply text-mining techniques[24–30] to 71 866 articles published in *The BMJ*

between 1948 and 2018 in order to find out what they say about women's health. Women's health was defined as health issues relating to biological characteristics (the female sex) or the behavioural, cultural or psychological traits typically associated with the female sex (gender). Historically, the difference between the usage of these terms has been less clear. We use the results of this analysis to provide a quantitative picture of the history of women's health, as seen through the lens of one journal, over the course of 70 years.

### Constructing a corpus of *The BMJ* research articles

The full-text *BMJ* corpus was constructed to explore the recent history of medicine.[26] The text was obtained by means of optical character recognition (OCR) on articles that were published from 1840 onwards and subsequently scanned. The meta-data and portable document formats (PDFs) are made publicly accessible by the US National Library of Medicine and PubMed Central (PMC). PMC records were used to verify the meta-data and retrieve electronic versions of full text starting from 2009. To ensure OCR quality, we used a subset of articles from 1 january 1948 to 18 July 2018 that had a digital object identifier (DOI) entry in the PMC database, were published in *The BMJ*, including the clinical research edition, and had a publication type of 'letter', 'article commentary', 'review article' or 'research article'.[27] Applying this criterion to 208 194 scanned articles and 2151 electronic articles resulted in 74 937 articles that had been published over the course of approximately 70 years. Figure 1 shows how these 74 937 research articles were filtered further to remove 1330 duplicate titles, 364 articles with fewer than 50 words and 749 articles not present in both the word

count and the topic probability datasets. In addition, we removed 628 articles published between 4 January 1997 and 14 February 1998 that were implausibly long, apparently due to parsing errors. The remaining 71 866 articles form the basis of our analysis.

### Topic modelling

We preprocessed the corpus by removing 571 common English words, stopwords using the Smart system,[31] and words which contained digits or hyphenated parts with less than one letter. To understand the content of the articles we used a text mining technique called topic modelling. The variant we used was Bayesian latent Dirichlet allocation (LDA)[32] implemented in the mallet-2.0.8 package.[33] The LDA algorithm estimates the probability of each word belonging to a topic and the probability of finding each topic in each article. This captures the likelihood of words appearing in the same document, but does not account for sentences, word order, capitalisation or punctuation. The number of topics, $k$, is set by the investigators; after some experimentation with $100 \leq k \leq 1000$, we settled on $k=400$. The Bayesian hyperparameters that control the dispersion of topics per article and words per topic were set at $\alpha=1/300$ and $\beta=1/100$; the algorithm optimised these every 10 epochs.

To aid the interpretation of our topics, a medical doctor (ENH) labelled them based on the 20 most probable words associated with each (see online data). A second reading was done by a coauthor with expertise in topic modelling (AML). Each topic was subsequently assigned to a group of related topics designated as a 'super topic'. For example, the super topic 'women breast cancer' contains three subsidiary topics: breast cancer, breast cysts (associated with cancer) and breast cancer screening. Since we were only interested in clinical medicine, we ignored the topics of healthcare management, the clinical literature, medical profession and regions. Of the 342 remaining topics that we used, 21 were clearly about women-specific health issues, such as pregnancy, oral contraceptives and breast cancer. Other topics besides these contained the word 'women' in their most-probable words, but since they also included the word 'men' they were not taken to be about women's health per se.

### Estimating the incidence of topics and words

Our LDA model produced a probability, $p_{ia}$, that the $i$th topic is found in the $a$th article. To simplify analysis topic probabilities were discretised, labelling a topic as present if it exceeds a threshold probability.[28] Specifically, we identified a set of topics of interest for a given analysis (eg, all clinical topics, or all medical condition topics), standardised the topic probabilities, $p_{ia}$ of each article by the summed probabilities of the topics in that set, and then discretised them by assuming that an article is 'about' some topic if $p_{ia} \geq 0.05$. Analogously, an article is about a given super topic if the summed probabilities of the subsidiary topics$\geq 0.05$. Having obtained counts of all

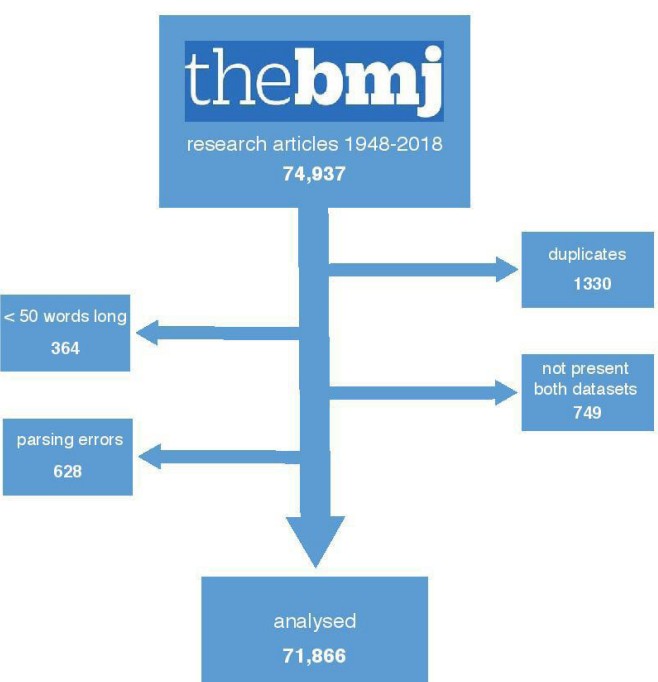

research articles 1948-2018
**74,937**

duplicates
1330

< 50 words long
364

not present
both datasets
749

parsing errors
628

analysed
**71,866**

**Figure 1** Constructing a corpus of *BMJ* research articles 1948–2018.

topics of interest in all papers, we estimated the incidence rate of a topic as:

$$I_{it} = \frac{N_{it}}{N_t}$$

where $N_{it}$ is the number of papers published at time $t$ that are about the $i$th topic, and $N_t$ is the total number of topics of interest identified in all papers published at that time. Note that, since papers may be about multiple topics, the denominator is not the total number of papers published in a given year. In the same way, we estimate the incidence of a word as the number of times it occurs relative to the count of all words in all articles published at time $t$.

## Constructing and analysing subcorpora

Here, we describe how we classified our articles into subcorpora.

### Articles about women-specific health topics and not

We classified articles into those that are about any aspect of women-specific health and those that are not. To do this, we first filtered our dataset for the 342 clinical topics. Then, for each article, we summed probabilities of all 21 women's health-specific topics, as well the summed probabilities of all the other topics. We then standardised these two probabilities by their sum, and applied the $p_t \geq 0.05$ threshold to the standardised probabilities. These steps identified 10 158 papers about women-specific health.

### Articles that are gender vocal and gender silent

We classified articles into those that are gender vocal and gender silent. To do this, we searched all articles that contained the words "woman/en" OR "female/s" OR "gender" OR "sex/es". Articles that used any of these words were labelled 'gender vocal', that is, differences between sex or gender, male or female, man or woman were evidently discerned, and those that used none of them 'gender silent'. We validated our classification by examining the full text of 100 articles—50 chosen at random from each group—and found that nearly all gender-vocal articles indeed studied women or girls. Gender silent articles, however, may study women or girls, but just not discuss them. These steps identified 28 837 gender-vocal articles.

### Statistical analysis

All analyses were done in R. To model word or topic incidence as a function of time, we used a binomial generalised linear model (GLM) implemented in base R or a generalised additive model implemented in the mcgv package. In order to identify breaks in individual series we carried out general linear regressions using the binomial family implemented in the segmented package[34] to estimate breakpoints. In all analyses we modelled time using publication date rather than year; for clarity, however, we show estimates of yearly incidence.

## Patient and public involvement

No patients were involved in setting the research question or the outcome measures, nor were they involved in the design and implementation of the study.

## RESULTS

### The rise of 'women' 1948–2018

To discover how the representation of women has changed over time in *The BMJ* we began by examining word counts. Considering all articles, the words 'woman' or 'women' appeared about three times in every thousand words (0.27%), however, this incidence increased from about one in a thousand in 1948 to about six in 2018 (figure 2A). Using a GLM, we estimated that the odds of a word being 'woman/women' increased by an annual factor of 1.023±0.0003 (estimate±95% CI); p<2.0 · $e^{-16}$, faster than 81% of the 1000 most frequent words of clinical relevance (figure 2B). By contrast, the odds of a word being 'man/men' increased by 1.002±0.0004; p<2.0 · $e^{-16}$)—about an order of magnitude slower.

We next asked whether the representation of women increased at the same rate in the articles of all clinical specialties. In fact, it is implausible that they should. This is because some specialties embrace important women-specific health conditions whose importance has varied over time. For example, the discovery in the 1960s that oral contraceptives and, later, hormonal replacement therapy, are risk factors for venous thrombosis resulted in many studies, quite a few of which were published in *The BMJ*.[35–44] In our scheme, they ephemerally increase the representation of women in articles about 'haematology'.

Here, however, we sought to exclude such effects. Instead, we wanted to study the frequency of occurrence of the words 'woman' and 'women' in articles that could, at least in principle, be about both sexes. For this reason, we first identified and excluded all articles about women-specific health topics (see below), which left us with 62 109, the bulk of the original sample. We then classified these articles into 20 clinical specialties using our super topics (eg, 'cardiology', 22 topics), and estimated the incidence of the words 'woman/women' in each using GLMs, as above. Figure 2C shows that the rate of increase of 'woman/women' varied considerably among these specialties. While the odds of a word being 'woman/women' increased by an annual factor of >1.02 in articles about nephrology, endocrinology and cardiology, many specialties had very low rates of increase (eg, pharmacology) or were indistinguishable from 0 (eg, anaesthesiology). One specialty, psychiatry, actually declined.

### The slow increase of women-specific health

Merely counting how often the words 'woman/women' were mentioned in *BMJ* articles does not, however, tell us what they are about. Topic analysis does. Of the 400 topics discovered by the LDA model, 21 were clearly about health issues specific to women. To capture whether an article is about any aspect of women-specific health we aggregated

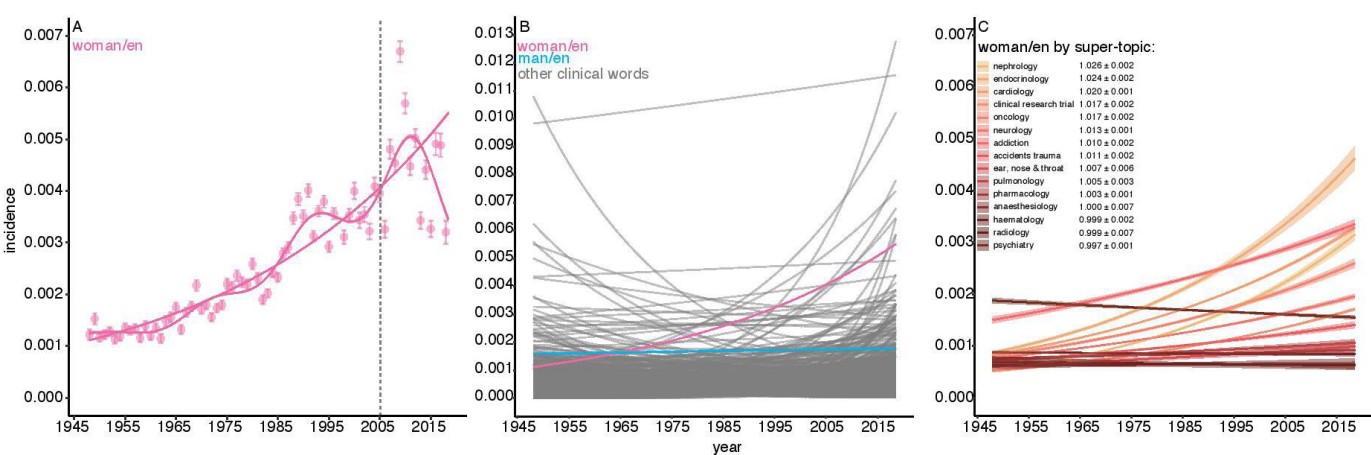

**Figure 2** The rise of 'women' in *The BMJ* 1948–2018. (A) The incidence of 'woman/women' by year; points are mean frequencies relative to all words, error bars are 95% CIs. Fitted lines are general additive and general linear models±1 SE based on publication date. The vertical grey dashed lines marks the date, March 2005, when Fiona Godlee became editor of *The BMJ*. (B) Predicted incidence of the 1000 most frequent words of clinical relevance, estimated by general linear models. 'Woman/women' is shown in pink; 'man/en' in blue; all others in grey. The rate of increase of 'woman/women', as estimated by the main date-of-publication effect of model fits, is faster than 81% of the other selected words. (C) The incidence of 'woman/women' by year in 20 clinical specialties defined by super-topics. The colour gradient indicates the rank order of the rate of increase; coefficient estimate (ORs±95% CI) of linear models for each specialty is given in the legend. The fastest increasing super topic is 'nephrology'; 'psychiatry' declines.

them into a single 'women-specific health' super topic. In 1948, the incidence of this supertopic was 11%; in 2018, 14%. The odds, then, that an article was about any aspect of women-specific health increased by an annual factor of $1.004\pm0.0013$; p=$9.28 \cdot e^{-10}$ (figure 3A).

This seems surprisingly slow. By way of comparison consider two other super topics: 'clinical trial reports'

(eight topics), which captures articles about randomised control trials, and 'clinical case reports' (five topics), which captures case studies. The odds that an article was about the first increased by an annual factor of $1.06\pm0.0018$; p<$2.0 \cdot e^{-16}$, and about the second decreased by an annual factor of $0.97\pm0.0008$; p<$2.0 \cdot e^{-16}$ (figure 3B). These super topics tell the great story of 20th century clinical science:

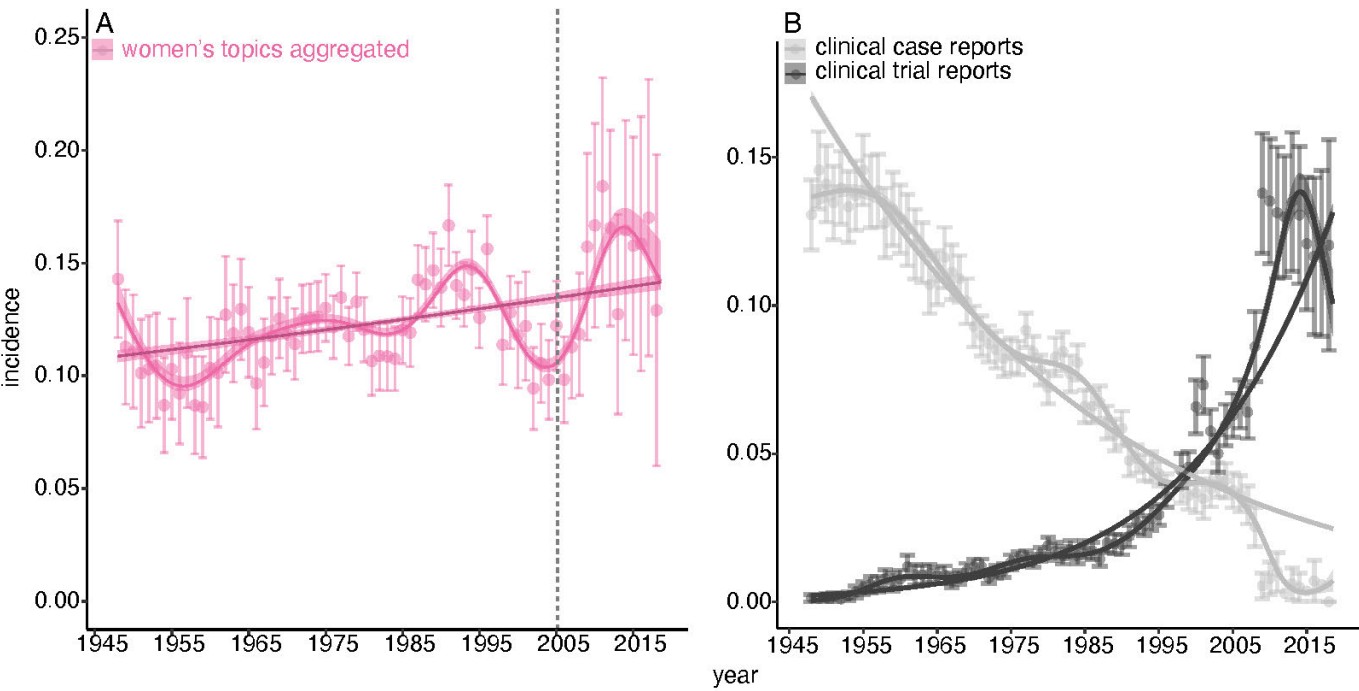

**Figure 3** Women-specific health in *The BMJ* 1948–2018. (A) The incidence of 21 aggregated women's health topics, error bars are 95% CIs; light pink lines is a general additive model, dark pink line is a single-break segmented general linear model: fits ±1 SE. The vertical grey dashed lines marks the date, March 2005, when Fiona Godlee became editor of *The BMJ*. (B) The rise of clinical trials and the decline of case studies. Point estimates are aggregated incidence of super topics, error bars are 95% CIs; lines are general additive model fits ±1 SE.

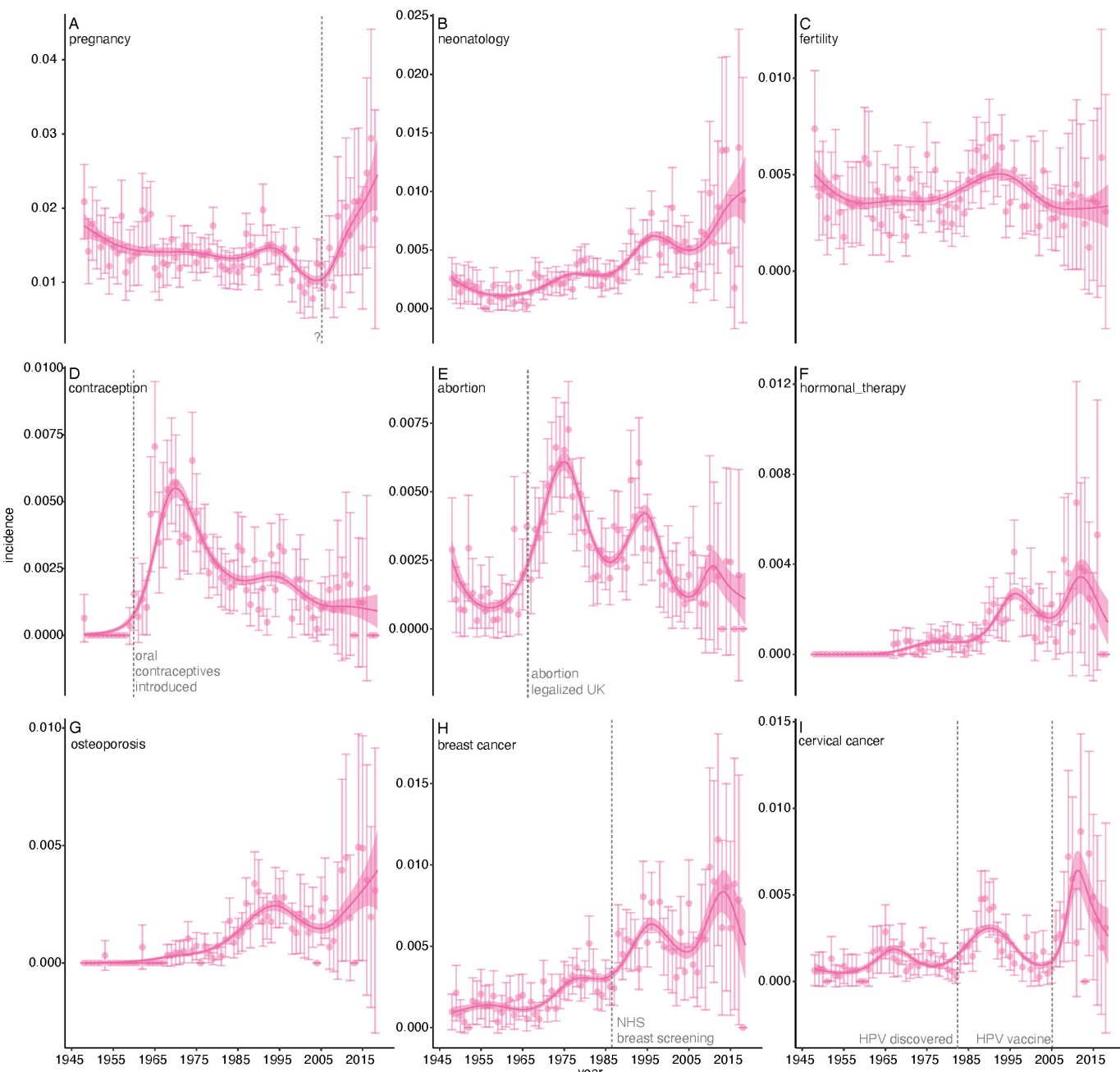

**Figure 4** Women's health topics in *The BMJ* research articles, 1948–2018. Point estimates are aggregated incidence of super topics, error bars are 95% CIs; lines are General additive model fits ±1 SE. Points are aggregated incidence of super topics, error bars are 95% CIs; lines are general additive model fits ±1 SE. Dotted vertical lines, and labels, mark major events in medicine that likely explain some of the major changes in super-topic incidence.

its transformation from case-based observations to testing hypotheses in groups of patients. The increase of women-specific health is a minor phenomenon compared with that—at least it is so as seen through the lens of *The BMJ*.

Why was the increase in articles about women-specific health so slow? To investigate this, we examined the subsidiary topics (figure 4). For clarity we grouped them into nine super topics: 'pregnancy' (8 topics),'neonatology' (2), 'fertility' (2, excluding male fertility), 'contraception' (1), 'abortion' (1), 'hormonal therapy' (1), 'osteoporosis' (1), 'breast cancer' (3) and 'cervical

cancer' (2). The most common women-specific super topic was 'pregnancy' (4744 articles); the least 'hormonal therapy' (319 articles). Fits of general additive models show that the historical dynamics of these super topics are characterised by fluctuations in incidence rather than a general increase. Not all changes are easily explained: the increase in articles about pregnancy that occurred around 2005 does not appear to be driven by any obvious medical breakthrough (figure 4A). But many of the largest fluctuations are easily explained. For example, the increase in studies about contraception after 1960 is associated with

the introduction of oral contraceptives[45] (figure 4D). The UK's Abortion Act of 1967,[46] engendered much discussion about that topic (figure 4E). The introduction of a breast screening programme by National Health Service in 1988 resulted in many articles discussing its efficacy (figure 4H). The 1983 discovery that the human papillomavirus's (HPV) caused cervical cancer prompted a surge of articles about that topic, as did the implementation of an HPV vaccine in 2006 (figure 4I). In general, it appears that the history of women-specific health reflects the impact of contingent events such as medical advances, public health programmes and sociolegal changes. Such events result in a flurry of articles which lasts for some years but then fades away as researchers turn to studying something else. The impact of the typical devaluation of topics that are feminised and the lack of women's representation in academic medicine have to be considered in this as well.

## It took a woman?

Besides estimating the incidence of the words 'woman/women', and the incidence of women's health-specific topics, in our articles, we also investigated how many of them mentioned women at all. To do this, we classified our articles into 'gender vocal' articles that discuss women—or at least mention gender and sex differences in some way—and 'gender silent' articles that are oblivious to them. We found that in 1948 around 40% (240/606) of articles were 'gender vocal', but that in 2018 84% (68/81) were (figure 5). In order to confirm our results, we read

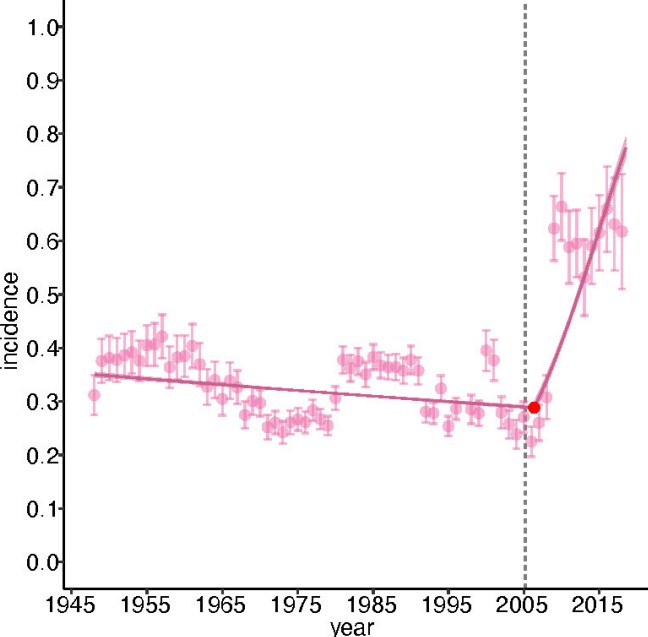

**Figure 5** The incidence of gender-vocal articles, 1948–2018. Error bars are 95% CIs; pink line is a single-break segmented general linear model that shows a strong structural break around 2005, when the incidence rapidly increased: fits±1 SE. The fit of this model is superior to one with no break by Akaike information criterion (AIC). The red point gives the breakpoint; the error bars (not visible) are 95% CIs.

the full text of many 'gender silent' papers published since 2010. We found some of them did, in fact, allude to women or sex or gender differences—but only in the data tables that we removed in preprocessing. This was particularly true of dozens of articles reporting clinical trials. Although these articles may have included women in their study samples, they were truly 'gender silent' in that they did not analyse or report or discuss sex-specific differences in any way.

The increase in 'gender vocal' papers did not occur gradually but rather suddenly, around 2005. A segmented GLM identified a single break at 2005±0.45 years (95% CI), before which the odds of an article being gender vocal had declined gradually by an annual factor of 0.99±0.0008; after which it increased by annual factor of 2.18±0.015. What was the cause of this dramatic change? One possibility is that it was due to a change in editorial policy. In March 2005, Fiona Godlee succeeded Kamran Abbas (acting, 2004–2005) and Richard Smith (1991–2004) to become the first female editor in chief in the journal's 180-year history. We estimate that an article published by her was about 31% more likely to contain the words 'woman/women' than one published by her immediate predecessors: Smith/Abbas: 0.0035±0.00003; Godlee: 0.0046±0.00005; ORs: 1.308±0.2949; $p<2 \cdot e^{-16}$. Similarly, an article published by her was 45% more likely to be gender-vocal than one published by her immediate predecessors: Smith/Abbas: 0.37±0.007; Godlee: 0.54±0.014; ORs: 2.04±0.13; $p<2 \cdot e^{-16}$. We did not detect a difference between the editors in the probability that an article is about women's health: Smith/Abbas: 0.0361±0.0012; Godlee: 0.0365±0.0023; ORs: 1.0134±0.0758; p=0.726). Thus, our data are consistent with the idea that Fiona Godlee had a substantial impact on the probability that a newly published article at least mentions women, but not on the probability that it considers some aspect of women-specific health.

## PRINCIPAL FINDINGS

We measured the representation of women in research articles in several ways. Two of these—the incidence of the words 'woman/women' relative to all words (figure 2A), and the incidence of gender-vocal articles (figure 5)—have quite different dynamics, but concur in showing that women are now much more likely to be discussed in research articles than in the past. For at least 50 years women have asked that clinical science treat them equally to men.[13–15 47] Clinical science, it seems, has responded—at least as seen in the pages of this journal, but the response has been uneven (figure 2C). When we excluded articles about health issues specific to women, and estimated the incidence of the words 'woman/women' in those that remained, we found that some—notably psychiatry—mentioned them relatively rarely; and that the rate at which they did so has not increased over seventy years. It is not that psychiatry has particularly many women-specific papers. We excluded only 9.4%

(615/6595) of psychiatry articles, and our topic analysis did not identify a single women-specific psychiatric topic such as postnatal depression. This contrasts with oncology in which we excluded 21.8% (1026/4711) of articles, mostly about breast and cervical cancer; nevertheless, the rate at which women were mentioned in the remaining articles improved substantially. Cardiologists, too, appear to be addressing their 'problem women'— as a *Lancet* editorial termed them[48]—but psychiatrists, it seems, must try harder.[49–51] Some are. The Royal College of Psychiatrists has had a 'Women and Mental Health' special interest group since 1995.

One contributing factor is that feminised topics, such as women's health, are devalued.[2] Researchers and perhaps also the audience or *The BMJ* itself may have less interest in women's health. Another factor is that women are more likely to focus on women's health issues and subsequent research, but possibly have been under-represented as authors for *The BMJ*. We set out to investigate this, but due to lack of bibliographic information, we were unable to assess the proportion of papers focusing on women's health to be published by female authors. Our finding that the focus on women and women's health appears to differ per discipline, could be a consequence of gender imbalance within the medical specialities. In general, gender imbalance within academic medicine and thus medical research have to be considered in interpreting these results.

Turning to articles about health issues specific to women, we found that their aggregated incidence has only increased very slightly over 70 years (figure 3A). Examination of the subsidiary super topics suggest that the dynamics of articles about women's health is driven less by a general trend towards increased representation, than by contingent medical advances, public health programmes and sociolegal events. We make no general claims as to whether or not women-specific health is, or has been, adequately represented in the journal's pages. But we can identify some absences. Topic analysis does so since it is an unsupervised machine learning method that depends on the coassociation of words among documents and, if a disease does not appear as a topic, that is probably because articles about it are rare. We have already mentioned one missing topic: postnatal depression. Endometriosis is another—even though it afflicts 1 in 10 women in their reproductive years.[52] Its relative invisibility is confirmed by simple word searches: 'pregnancy', 'breast', 'ovarian' appear in 366, 270 and 63 of 2305 articles published since 2008, but 'endometriosis' in only 15. These results are consistent with claims that the disorder has been neglected by clinical researchers and poorly understood by physicians.[53–55] Similarly, although women are far more likely to be the victims of domestic abuse than men,[56] its medical consequences did not appear as a topic in any form. By contrast, 'head injuries', which is mostly about concussions in rugby players, did. Such arguments cut both ways. The UK diagnosis and mortality rates of breast and prostate cancer are nearly the same,[57] but where breast cancer gets three topics ('breast cancer', 'breast cysts', and 'breast cancer treatment'), prostate cancer gets none. Word counts, however, show that the representation of these diseases is rapidly becoming more even (data not shown). Were we to repeat the analysis a decade hence, prostate cancer would likely have a topic of its own.

Our most intriguing result is the revelation that the incidence of gender-vocal articles increased dramatically around 2005, just when Fiona Godlee became editor in chief. It is easy to see how the two events might be causally connected. Perhaps authors, aware that *The BMJ* had acquired a new female editor, made sure to discuss the implications of their results in the context of women's health while previously they had been less assiduous in doing so. Or perhaps Godlee just made sure they did. Our results also suggest that her editorial intervention, if it existed, was limited, for we find no evidence that her editorship had an impact on whether or not an article was about a women-specific health topic. In order to clarify the interpretation of these results, we have written to Fiona Godlee asking her to comment on them. Additionally, in response to a lack of sex disaggregated data, some journals now require all data to be disaggregated by sex.

The effect that women editors in chiefs might have on the careers of other women has been much discussed.[58–63] As far as we know their effect on the articles that their journals publish has not. In the absence of direct evidence, we cannot exclude the possibility that the sudden rise of gender-vocality at the *The BMJ* is due a change in the sensibilities of its authors quite independent of its editor. Even if there is a causal association, we cannot say how general it might be. But we could, in principle, find out. Several other important medical journals—*The Lancet*, *JAMA*, *JAMA Internal Medicine*, *Annals of Internal Medicine* and the *Cochrane Library* among others—currently have, or have had, women editors in chief. Given the full texts of all their articles, it would not be difficult to apply our methods to those journals—as well as to the many that have never had one.

### Strengths and limitations of this study

Most histories of women's health are about pre-20th century medicine and rest on limited textual evidence.[9–12] Our study, by contrast, is about modern medicine, uses text mining tools to quantify the content of thousands of clinical articles, and inferential statistics to test hypotheses. The main limitation of our study is that it is based on a single journal, one that strongly reflects British concerns. Another is that topic analysis is a rather blunt instrument for discovering what articles are about.[27] Finally, statistical patterns of historical change are, by themselves, difficult to interpret in causal terms. But they should be of interest to historians who can investigate their causes using traditional methods

## Conclusions, recommendations and future directions

In the future, we hope to discover whether our findings can be generalised to the rest of the clinical literature. It would be particularly interesting to study other general medical journals. More sophisticated text mining tools might also allow the content of articles to be explored in a more nuanced way.[64 65] We strongly encourage all journals to require all data be disaggregated by sex and aim to maintain a balance in topics with attention for sex-specific differences. Finally, we note that we have made our code and data public (https://github.com/Armand1/Women-in-the-BMJ-public). Many other medical subjects might be studied much as we have studied women's health, and we invite others to do so.

**Acknowledgements** We thank Theodora Bloom, Sean Harrop and Josie Breen of The BMJ for sharing article meta data and Clare Isacke for comments on the manuscript.

**Contributors** ENH, SM and AML designed the study. AJB, SA and AML acquired the data. AJB and AML performed the analyses. ENH, JDK and AML drafted the manuscript. All authors provided critical comment and feedback on the manuscript. The corresponding author attests that all listed authors meet authorship criteria and that no others meeting the criteria have been omitted. SM and AML are the guarantors of this manuscript.

**Funding** The authors have not declared a specific grant for this research from any funding agency in the public, commercial or not-for-profit sectors.

**Competing interests** None declared.

**Patient and public involvement** Patients and/or the public were not involved in the design, or conduct, or reporting, or dissemination plans of this research.

**Patient consent for publication** Not required.

**Provenance and peer review** Not commissioned; externally peer reviewed.

**Data availability statement** Data are available in a public, open access repository. We have made our code and data public: https://github.com/Armand1/Women-in-the-BMJ-public.

**ORCID iD**
Eva N Hamulyák http://orcid.org/0000-0002-9340-8771

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
