## [Reviewer comments · BMJ Open]

ARTICLE DETAILS

TITLE (PROVISIONAL)	Women's Health in The BMJ: A Data Science History
AUTHORS	Hamulyák, Eva; Brockmeier, Austin; Killas, Johanna; Ananiadou, S; Middeldorp, Saskia; Leroi, Armand

VERSION 1 – REVIEW

REVIEWER	Rosemary Morgan Johns Hopkins Bloomberg School of Public Health, USA
REVIEW RETURNED	21-Jun-2020

GENERAL COMMENTS	This is a very interesting paper focusing on an important topic. I enjoyed reading it. There are a few areas that could be strengthened throughout the paper which would further help to contextualize the topic. The findings support gender bias which are prevalent within academic medicine. There are some key gender-related issues that are not mentioned, including women's lack of representation within leadership and medicine throughout the years, the lack of sex disaggregated data, and the fact that topics which are feminized are typically devalued. Contextualizing the findings in relation to some of these issues would strengthen the paper. Currently the introduction needs to be expanded to provide appropriate background and context. The introduction currently does not provide enough justification for why this was an important topic to study, what the main implications are, or why should researchers and policymakers care about what articles in the journal said about women's health and how this has changed over time, etc. Some areas that could be expanded include: • Page 3, lines 11-15 it states: "Compared to men, women have historically been and, in some ways still are, ill-served by medicine [5]. The deficiencies of early medicine in the treatment of women are well documented [9–12], and so are the twentieth century's campaigns for change [13–17]." It is possible that people reading your article will not be familiar with how women have been ill-served by medicine. The paper would be strengthened by providing some context and examples. What are the deficiencies of early medicine in the treatment of women, how is this reflected in contemporary times, what are the 20th century's campaigns for change?• Page 3, lines 20-21 states: "But the question of how women have been studied by clinical science is a much larger one than this. And its answer has consequences for health policy [1]." Please elaborate. What do you mean by "the question of how women have been studied by clinical science is a much larger one than
---

	this.” How so? In what way? What are the consequences for health policy? I would like to see some discussion in the introduction about women’s health in relation to sex and gender (and how gender does not equal women’s health – this is important due to search terms you used). How are you defining women’s health? How are you defining gender? How the authors are approaching sex and gender needs more explanation as as it is currently written it seems that the two are being conflated which is incorrect. Introduction needs to provide context about why you chose to focus on clinical medicine only. Page 5, lines 27-32 states” We classified articles into those that are gender vocal and gender silent. To do this we searched all articles for words that are related with gender or sexual differentiation: “woman/en”, “female”, “gender” and “sex”, and labelled articles that used any of these words “gender-vocal”, and those that used none of them “gender-silent”. Did you look for women OR female OR gender OR sex? Or women OR female AND gender OR sex? Or did you look for sex and gender only after you searched for women and female? This is important as gender or sex does not necessarily equate with women and it is important that this is not perpetuated in this paper. You can’t assume that something that focused on gender was focused on women or something that focused on women also took a gender lens. This is a common misconception that is often perpetuated within the literature. You state that you “found that nearly all gender-vocal articles indeed studied women or girls” – this is evidence to the fact that gender is often conflated with women and girls or used specifically for studies focusing on women and girls. Providing clear definitions about sex and gender will help. If you are defining gender as women make this clear and state that this not in align with how gender is (or typically should be) defined. Also – please provide more information about what you mean by gender vocal and gender silent. These terms are not common and need to be explained. Page 7, lines 42-43 states: “But, in general, it appears that the history of women-specific health reflects the impact of contingent events such as medical advances, public health programmes, and socio-legal changes.” Slow rate of increase in focus on women’s health topics could reflect a bias in academic medicine or the BMJ itself against women’s health. Topics which are feminized, such as women’s health are typically devalued. As such, they are seen as less important or not “real science”. The lack of women’s representation in academic medicine and related health disciplines (as women are more likely to focus on women’s health issues) also impacts this. It would be interesting to know how many of the articles that focused on women’s health were published by women. Your findings related to the increase in gender vocal articles after Fiona Godlee are very interesting and provide good justification for the need for gender policy within editorial boards and diverse representation of editors. This also supports the argument that
--	---

	women put more importance on women's health and gender issues. Your finding that certain disciplines focus on women and women's health less than others, such as psychiatry, may also be a reflection of the fact that these disciplines are dominated by men. It would be good to see some discussion of this within the paper. The findings likely also reflect a lack of sex disaggregated data overall – something which has been written about a lot. Sex differences disappear when data is aggregated. Some journals are now requiring that all data be disaggregated by sex. I would like to see this as a key recommendation coming out of the paper.
--	---

REVIEWER	Marc Lerchenmueller University of Mannheim Germany
REVIEW RETURNED	26-Jun-2020

GENERAL COMMENTS	In the manuscript "Women's Health in The BMJ: A Data Science History" (bmjopen-2020-039759) the authors present interesting and valuable research on the historical development of gender coverage in medical research articles published in the BMJ. To compare subdisciplines and to identify different kinds of articles the authors employ topic modelling. During the time period of 1948 to 2018 the authors find a steady increase of papers mentioning the words "woman" or "women". For articles covering women-specific health they do not detect such a clear linear pattern but rather fluctuations that they attribute to medical advances or socio-legal changes. For articles, which allude to some aspect of gender (what the authors call gender-vocal) there seems to be a steady decrease from 1948-2005 followed by a particularly sharp increase shortly after Fiona Godlee became the first female editor of the BMJ. Major point(s) The authors present a number of results, which are all interesting in and of themselves but come with different implications: i) clinical studies replace case studies, ii) articles with the words "woman/en" are published more, iii) articles covering women health come and go with societal changes, iv) gender vocal articles increase with a female editor. I would encourage the authors to focus on the ones that they deem most relevant to the problem they focus on, i.e. the representation of the female gender in medical research. There remain some concerns on how much of the sharp increase in "gender vocal" papers around 2005 can really be attributed to Fiona Godlee. The relatively larger standard errors in estimates relating to more recent publications as well as some numbers mentioned in the text (e.g., p.9 line 52) suggest that there is a lot fewer articles from the mid-2000s onwards as compared to the years/ decades before. It is not clear whether this drop in the number of publications can be attributed to changes in editorial policies or, for example, to how the databases record and index publication types, which was one of the filters used in the creation of the dataset. I would therefore encourage the authors to share more descriptive statistics on article characteristics and publication policies at BMJ and how they have changed over the time under study (e.g., # articles per year/decade, av. length of article/ decade, # topics per article/ decade, publication types/ decade). If,
--

	for instance, the BMJ were to publish fewer but longer research articles in more recent years, there would be more page space to discuss gender, even if the paper itself is not focused on gender-specific health. Also, it might be more likely that women-specific health topics are detected in the text corpus of a long article. Minor point(s) It would add to the robustness of results if the authors could run a sensitivity analysis with different thresholds for topic assignment, especially since many of the topics seem to share at least some of their most frequent words (e.g., the word “patient” is the most frequent word in 16 different topics). The authors define gender-vocal articles as articles containing the words “woman/en”, “female”, “gender”, and “sex”. To me it is not obvious why the words “man/en”, “male” and also the plural forms of “females, sexes, males” are not included in this list of words as they would also reflect some sort of discussion of gender. Particularly as clinical trials replace case studies plural forms might become even more relevant. It seems that topics were assigned by one medical expert. It would be nice to have at least a second expert define topics and to calculate the inter-rater reliability. As the authors point out, topic modelling is a neat explorative method but cannot sharply identify disciplines. In a subsample analysis, the authors might also consider comparing their identified topics to the MeSH terms associated with specific articles. It seems the p-value on p. 9, line 25 is wrongly reported (1.308 ± 0.2949; $P < 2 \times 10^{-16}$.) Depending on how regularly the BMJ publishes special issues the content of such issues might explain some of the presented results. Maybe the authors can discuss and look at the influence of such issues, e.g., by excluding them from some of the analyses.
--	---

VERSION 1 – AUTHOR RESPONSE

Reviewer: 1

Reviewer Name: Rosemary Morgan

Institution and Country: Johns Hopkins Bloomberg School of Public Health, USA

Please state any competing interests or state ‘None declared’: None declared.

Please leave your comments for the authors below

This is a very interesting paper focusing on an important topic. I enjoyed reading it. There are a few areas that could be strengthened throughout the paper which would further help to contextualize the topic. The findings support gender bias which are prevalent within academic medicine. There are some key gender-related issues that are not mentioned, including women’s lack of representation within leadership and medicine throughout the years, the lack of sex disaggregated data, and the fact that topics which are feminized are typically devalued. Contextualizing the findings in relation to some of these issues would strengthen the paper.

We thank the reviewer for these kind words and excellent suggestions for improvement. These have now been incorporated in the Introduction section: “An unequal representation of women exists in

leadership and medicine, and women-specific topics are often devalued [2].”

By means of the gender-vocal/gender-silent analysis, we aimed to explore whether studies analysed existing differences between sexes and whether or not possible differences were explicitly discussed. In general, journals should require all data be disaggregated by sex. We have added the following lines to the Discussion section of the paper to address a general lack of sex disaggregated data and as one of our recommendations: “Additionally, in response to a lack of sex disaggregated data, some journals now require all data to be disaggregated by sex”.

In the Conclusions, recommendations, and future directions section: “We strongly encourage all journals to require all data to be disaggregated by sex and aim to maintain a balance in topics with attention for sex-specific differences.”

Currently the introduction needs to be expanded to provide appropriate background and context. The introduction currently does not provide enough justification for why this was an important topic to study, what the main implications are, or why should researchers and policymakers care about what articles in the journal said about women’s health and how this has changed over time, etc.

Some areas that could be expanded include:

- Page 3, lines 11-15 it states: “Compared to men, women have historically been and, in some ways still are, ill-served by medicine [5]. The deficiencies of early medicine in the treatment of women are well documented [9–12], and so are the twentieth century’s campaigns for change [13–17].” It is possible that people reading your article will not be familiar with how women have been ill-served by medicine. The paper would be strengthened by providing some context and examples.

We thank the reviewer for this suggestion. We have now provided some context and examples, by adding the following lines to the Introduction section: “The tendency to use men as the standard in clinical research, driven by concerns for potential teratogenic effects of drugs or by deeming women’s inclusion as riskier, have been suggested as explanations for the underrepresentation of women in trials [5]. Even if both sexes were included, sex disaggregation was not performed, as it is becoming standard now [21].

Clearly, sex-specific differences are present and must be considered in clinical-decision making, but the question of how women have been studied by clinical science is a much larger one than this.”

- Page 3, lines 20-21 states: “But the question of how women have been studied by clinical science is a much larger one than this. And its answer has consequences for health policy [1].” Please elaborate. What do you mean by “the question of how women have been studied by clinical science is a much larger one than this.” How so? In what way? What are the consequences for health policy?

The way women have been studied by clinical science is expressed in representation in clinical trials, attention for women’s health issues and acknowledgement of sex-specific differences. But perhaps even more important, what has been learned from this research, how much has been put into practice or communicated to health care providers or the women themselves, i.e. consequences for health policy. We have now added the following statement: “...but the question of how women have been studied by clinical science is a much larger one than this. It includes the focus and emphasis of medical research, the communication these findings and their translation to clinical practice, all of which have consequences for health policy [1].”

I would like to see some discussion in the introduction about women's health in relation to sex and gender (and how gender does not equal women's health – this is important due to search terms you used). How are you defining women's health? How are you defining gender? How the authors are approaching sex and gender needs more explanation as as it is currently written it seems that the two are being conflated which is incorrect.

We agree with the reviewer that our approach to “sex”, “gender” and “women's health” requires further defining in context of the current study. Indeed, “sex” and “gender” should not be conflated. However, in the past these terms have been conflated and given our analysis of 70 years history of published BMJ articles, the retrospective nature of our research, we needed to take a broader view. We do acknowledge the importance of explicitly stating this and have now added the following definition of women's health we used in the current study in the Introduction section of the paper, as well as a statement on the historic usages of the aforementioned terms.

“Women's health was defined as health issues relating to biological characteristics (the female sex) or the behavioural, cultural or psychological traits typically associated with the female sex (gender). Historically, the difference between the usage of these terms has been less clear.”

Introduction needs to provide context about why you chose to focus on clinical medicine only.

We thank the reviewer for this suggestion. We chose to focus on clinical medicine, as we were interested in how women and women's health were studied and subsequently published in a clinical journal like The BMJ in a set period of time. We have now added the issues with the representation and role of women in for instance health care management or medical training in the introduction, as it is important albeit not the focus of our research.

Page 5, lines 27-32 states” We classified articles into those that are gender vocal and gender silent. To do this we searched all articles for words that are related with gender or sexual differentiation: “woman/en”, “female”, “gender” and “sex”, and labelled articles that used any of these words “gender-vocal”, and those that used none of them “gender-silent”. Did you look for women OR female OR gender OR sex? Or women OR female AND gender OR sex? Or did you look for sex and gender only after you searched for women and female? This is important as gender or sex does not necessarily equate with women and it is important that this is not perpetuated in this paper. You can't assume that something that focused on gender was focused on women or something that focused on women also took a gender lens. This is a common misconception that is often perpetuated within the literature. You state that you “found that nearly all gender-vocal articles indeed studied women or girls” – this is evidence to the fact that gender is often conflated with women and girls or used specifically for studies focusing on women and girls. Providing clear definitions about sex and gender will help. If you are defining gender as women make this clear and state that this not in align with how gender is (or typically should be) defined.

We thank you for this important remark and we agree that this requires clarification in the manuscript. The Introduction section now contains clear definitions of women's health, sex and gender used in the current study. We looked for “woman/en” OR “female” OR “gender” OR “sex” and have now rephrased accordingly. We agree that gender or sex do not necessarily equate with women. However, with this analysis we aimed to evaluate whether or not articles discerned different sex or gender, not whether the article focused on women per se. For instance, an article containing the sentence

“Incidence of bleeding events did not differ between the sexes”, would have been classified as gender-vocal, since it tells us that the incidence was analysed separately for male and female sex.

Also – please provide more information about what you mean by gender vocal and gender silent. These terms are not common and need to be explained. We have added the following clarification to this section:

“To do this we searched all articles that contained the words “woman/en” OR “female/s” OR “gender” OR “sex/es”. Articles that used any of these words were labelled “gender-vocal”, i.e. differences between sex or gender, male or female, man or woman were evidently discerned, and those that used none of them “gender-silent”.

Page 7, lines 42-43 states: “But, in general, it appears that the history of women-specific health reflects the impact of contingent events such as medical advances, public health programmes, and socio-legal changes.” Slow rate of increase in focus on women’s health topics could reflect a bias in academic medicine or the BMJ itself against women’s health. Topics which are feminized, such as women’s health are typically devalued. As such, they are seen as less important or not “real science”. The lack of women’s representation in academic medicine and related health disciplines (as women are more likely to focus on women’s health issues) also impacts this.

It appears many of the largest observed fluctuations in Figure 4 can be directly linked to certain medical advances and socio-legal changes, with women’s health overall slowly increasing over time. The lack of women’s representation in academic medicine and related health disciplines could definitely induce bias in academic medicine. We think this could have had an impact on the rate of increase in focus on women’s health, certainly not expediting it over the years. We do however wonder whether, unfortunately, this has been a relatively constant factor over time.

We have added the following in the Results section to address these other factors likely affecting the slow rate of increase in women’s health: “The impact of the typical devaluation of topics that are feminized and the lack of women’s representation in academic medicine have to be considered in this as well.”

Additionally, we now touch upon this in the Discussion section as well: “One contributing factor is that feminized topics, such as women’s health, are devalued. Researchers and perhaps also the audience or The BMJ itself may have less interest in women’s health.”

It would be interesting to know how many of the articles that focused on women’s health were published by women.

This is an excellent suggestion and indeed, there is a lot of evidence that women publish less as senior and so on. We did set out to look into this, but unfortunately due to lack of bibliographic information we were unable to determine the sex of the authors. We have added this statement to the Principal findings section of the paper.

“Another factor is that women are more likely to focus on women’s health issues and subsequent research, but possibly have been underrepresented as authors for The BMJ. We set out to investigate this, but due to lack of bibliographic information, we were unable to assess the proportion of papers

focussing on women's health to be published by female authors."

Your findings related to the increase in gender vocal articles after Fiona Godlee are very interesting and provide good justification for the need for gender policy within editorial boards and diverse representation of editors. This also supports the argument that women put more importance on women's health and gender issues.

Thank you, we further elaborate on this in the Discussion section.

Your finding that certain disciplines focus on women and women's health less than others, such as psychiatry, may also be a reflection of the fact that these disciplines are dominated by men. It would be good to see some discussion of this within the paper.

We thank the reviewer for this suggestion. We have incorporated this suggestion into the Discussion section of the paper: "Our finding that the focus on women and women's health issues appears to differ per discipline, could be a consequence of gender imbalance within the medical specialties. In general, gender imbalance within academic medicine and thus medical research have to be considered in interpreting these results."

The findings likely also reflect a lack of sex disaggregated data overall – something which has been written about a lot. Sex differences disappear when data is aggregated. Some journals are now requiring that all data be disaggregated by sex. I would like to see this as a key recommendation coming out of the paper.

We thank the reviewer for this suggestion and have incorporated this as a recommendation.

Reviewer: 2

Reviewer Name: Marc Lerchenmueller

Institution and Country: University of Mannheim, Germany

Please state any competing interests or state 'None declared': None declared

Please leave your comments for the authors below

In the manuscript "Women's Health in The BMJ: A Data Science History" (bmjopen-2020-039759) the authors present interesting and valuable research on the historical development of gender coverage in medical research articles published in the BMJ. To compare subdisciplines and to identify different kinds of articles the authors employ topic modelling. During the time period of 1948 to 2018 the authors find a steady increase of papers mentioning the words "woman" or "women". For articles covering women-specific health they do not detect such a clear linear pattern but rather fluctuations that they attribute to medical advances or socio-legal changes. For articles, which allude to some aspect of gender (what the authors call gender-vocal) there seems to be a steady decrease from 1948-2005 followed by a particularly sharp increase shortly after Fiona Godlee became the first female editor of the BMJ.

Major point(s)

The authors present a number of results, which are all interesting in and of themselves but come with

different implications: i) clinical studies replace case studies, ii) articles with the words “woman/en” are published more, iii) articles covering women health come and go with societal changes, iv) gender vocal articles increase with a female editor. I would encourage the authors to focus on the ones that they deem most relevant to the problem they focus on, i.e. the representation of the female gender in medical research.

We thank the reviewer for this suggestion. We measured the representation of women in research articles in several ways, focussing on 1) the incidence of woman/en and women’s health topics over time and 2) the incidence of gender-vocal articles. We aimed to put the evolution of women-specific health in perspective by way of comparison and therefore chose to briefly mention another supertopic (clinical studies replacing case studies) or women’s health in relation to societal changes. Although not our main focus, we think this best illustrates women’s health evolving over time, as seen through the lens of The BMJ.

There remain some concerns on how much of the sharp increase in “gender vocal” papers around 2005 can really be attributed to Fiona Godlee. The relatively larger standard errors in estimates relating to more recent publications as well as some numbers mentioned in the text (e.g., p.9 line 52) suggest that there is a lot fewer articles from the mid-2000s onwards as compared to the years/decades before. It is not clear whether this drop in the number of publications can be attributed to changes in editorial policies or, for example, to how the databases record and index publication types, which was one of the filters used in the creation of the dataset. I would therefore encourage the authors to share more descriptive statistics on article characteristics and publication policies at BMJ and how they have changed over the time under study (e.g., # articles per year/decade, av. length of article/ decade, # topics per article/ decade, publication types/ decade). If, for instance, the BMJ were to publish fewer but longer research articles in more recent years, there would be more page space to discuss gender, even if the paper itself is not focused on gender-specific health. Also, it might be more likely that women-specific health topics are detected in the text corpus of a long article.

We thank the reviewer for raising this interesting point. Our main aim was to determine how the representation of women’s health has changed in clinical studies using a corpus of BMJ articles published in the past 70 years. Even if page space was directly correlated with gender being discussed, attention is warranted and that is what we aimed to do by conducting this investigation, highlight the equal opportunity between men and women for being researched and discussed. In constructing the corpus of BMJ articles, we did look into changing publication policies and article characteristics over time. We evaluated whether paper length and publication date should be included as a covariate in our model and found the effects to be so small compared to the main effect, that we decided to neglect these. We have made our code and data public (<https://github.com/Armand1/Women-in-the-BMJ-public>), and chose to present the specific data on article characteristics and publication policies online.

Minor point(s)

It would add to the robustness of results if the authors could run a sensitivity analysis with different thresholds for topic assignment, especially since many of the topics seem to share at least some of their most frequent words (e.g., the word “patient” is the most frequent word in 16 different topics).

We thank the reviewer for this suggestion. After some experimentation with the number of topics, we settled on $k = 400$. We found that an increase to $k = 1000$ did not give us more granularity regarding the topics. In general, the words are not meant to be unique or identified with one particular topic, but

rather reflect the conditional probability of the word given the topic. This is best illustrated by “patient” being the most frequent word in a corpus of articles on clinical studies.

The authors define gender-vocal articles as articles containing the words “woman/en”, “female”, “gender”, and “sex”. To me it is not obvious why the words “man/en”, “male” and also the plural forms of “females, sexes, males” are not included in this list of words as they would also reflect some sort of discussion of gender. Particularly as clinical trials replace case studies plural forms might become even more relevant.

Plural forms were picked up as well and we have now clarified this in the Methods section on page 5: “To do this we searched all articles that contained the words “woman/en” OR “female/s” OR “gender” OR “sex/es”.”

It seems that topics were assigned by one medical expert. It would be nice to have at least a second expert define topics and to calculate the inter-rater reliability. As the authors point out, topic modelling is a neat explorative method but cannot sharply identify disciplines. In a subsample analysis, the authors might also consider comparing their identified topics to the MeSH terms associated with specific articles.

We thank the reviewer for pointing this out. All topics were assigned by one medical expert (EH), while a second reading was done by a co-author with expertise in topic modelling(AL). We have now added this to the Methods section. Inter-rater reliability would definitely have been interesting. However, the topic labels were open-choice, not predefined upfront, which would complicate this assessment and therefore be of limited value. A subsample analysis comparing our identified topics to the MeSH terms would be very interesting, but we considered this to be beyond the scope of the current manuscript.

It seems the p-value on p. 9, line 25 is wrongly reported (1.308 ± 0.2949 ; $P < 2 \times 10^{-16}$.)

We have double checked this and the p-value is in fact correct, most likely a consequence of the current sample size of 71.000 for these computations.

Depending on how regularly the BMJ publishes special issues the content of such issues might explain some of the presented results. Maybe the authors can discuss and look at the influence of such issues, e.g., by excluding them from some of the analyses.

This is certainly an interesting suggestion. Given that the BMJ is weekly peer-reviewed journal and the much lower frequency of special issues, such as the BMJ Christmas or BMJ ABC editions, being published, we are unsure whether this would have an effect on our results. Unfortunately, we did not formally test this hypothesis, as we had to make choices what to present in the current manuscript.

VERSION 2 – REVIEW

REVIEWER	Rosemary Morgan Johns Hopkins Bloomberg School of Public Health, USA
REVIEW RETURNED	01-Sep-2020

GENERAL COMMENTS	The authors have effectively addressed all comments.
--

REVIEWER	Marc Lerchenmueller University of Mannheim, Germany
REVIEW RETURNED	11-Sep-2020

GENERAL COMMENTS	The authors have revised the presentation of their manuscript in response to the comments provided. Certain recommendations were viewed as out of scope by the authors and the decision here lies with the editor. Thank you for the opportunity to review this interesting and important piece of research.
---